# Graph reinforcement learning resistant to sparsity scaling

## Abstract

We investigate the impact of graph sparsity on the NP-hard combinatorial optimization problem of delivery route optimization, by combining proximity policy optimization with graph convolutional neural networks. Sparsity poses a critical challenge for consistent routing policies, as limited connectivity can significantly impact solution strategies by changing the intrinsic structure of possible paths. In order to address such challenge, we relate robustness in different graph sparsity regimes to learning dynamics in a sequential decision-making and graph-structured environment. Our experiments systematically consider graphs with up to 20 nodes, demonstrating the robustness of the algorithm to various densities of graph sparsity, from low-degree nodes that impose inefficient paths to highly connected structures that require extensive exploration. To ensure consistent evaluation across different graph topologies, we introduce the normalization of the return function based on the length of the DRL episode and the number of nodes in the graph. The algorithm is evaluated based on metrics such as cumulative reward, path length, and the number of steps required to complete a DRL episode. Learning maintains stable performance across a wide range of graph densities, thus revealing its effectiveness. By systematically characterizing the role of sparsity in graph-based reinforcement learning for route optimization, this work provides insights into the challenges posed by real-world transportation logistics networks.

## 1 Introduction

We implement and evaluate a graph reinforcement learning framework that addresses routing optimization at different levels of sparsity and scale. The proposed approach combines proximity policy optimization (PPO) (Schulman et al., 2017) with a graph convolutional neural network (GCNN) (Wu et al., 2021), enabling the agent to directly tackle a graph-structured environment. We aim at proving the robustness of this framework in determining the goods delivery strategy for a driver within a randomly generated network of routes, by varying the sparsity regime. Such topology reflects the real world delivery networks, typically sparse and irregular, so the evaluation of algorithmic robustness under variable connectivity is critical to ensuring reliable routing performance in practical applications.

Many combinatorial optimization and routing problems can be naturally considered as sequential decision-making problems formulated on graphs. Therefore, they can be effectively addressed using deep reinforcement learning (DRL) methods (Sutton & Barto, 2018; Li, 2017), such as PPO (Lazaridis et al., 2020; Yimer et al., 2025), combined with graph neural networks (Zhang et al., 2019), which leads to Graph Reinforcement Learning (GRL) (Munikoti et al., 2023; Nie et al., 2023). Notable applications of GRL include the traveling salesman problem (TSP) and related routing and scheduling problems (Almasan et al., 2022; Kool et al., 2018; Kwon et al., 2020; Chen et al., 2022; He et al., 2024; Fellek et al., 2024; Pan et al., 2023; Park et al., 2021). However, most existing GRL approaches assume dense or fully connected graphs, limiting their relevance to real-world networks. In practice, transportation and logistics networks are sparse and irregular, where connectivity constraints critically affect routing efficiency and robustness. Building on previous work in which DRL has been successfully applied to a wide range of physics and engineering problems (Cestero et al., 2022; Parisi et al., 2016; Moro et al., 2021; Semola et al., 2022; Porotti et al., 2019; Corli et al., 2021; 2023), the present study focuses on combinatorial optimization on random and sparse graphs, addressing a more realistic and less explored context in the current literature on GRL. Our approach leverages a GCNN

that preserves node features and graph connectivity, forming the neural core of the GRL agent to effectively model routing dynamics of delivery drivers. Since real-world delivery networks are not fixed, we model the problem as an environment that requires sequential decision-making that can be generalized to changing and sparse instances, where simple heuristics like greedy algorithms are ineffective. We evaluate the agent up to 20 nodes and show that this heuristic can be robust with various levels of graph sparsity. In particular, it ranges from sparse graphs with numerous low-degree nodes, leading to suboptimal path traversal, to highly connected topologies requiring extensive exploration due to multiple routing options. To facilitate a fair comparison between different topologies, we normalized the return function by incorporating both the length of the episode and the number of nodes. Agent training performance is validated using metrics such as cumulative reward, path length, and number of steps required to complete an RL episode. By addressing routing problems with varying degrees of graph sparsity, our work establishes practical insights into the generalization and robustness of GRL methods in realistic network contexts, providing a valuable benchmark for future research on graph optimization. The work is structured into three sections. The first Section explains the preliminary concept of graphs data, GNNs and a brief recap of PPO. The second Section describes the modeling of the problem and describes the agent architectures. The third Section shows the results of applying the method to solve the delivery path problem. The final Section summarizes the results obtained and proposes conclusions.

## 2 Preliminaries

This preliminary Section begins with defining a graph, an abstract mathematical object suitable for describing systems composed of several interacting units, provides the basis of GNNs, and then introduces the PPO algorithm.

### 2.1 Basics of graph data

A graph is an abstract mathematical structure that models pairwise relationships between objects. Formally, a graph $G$ is defined as a pair $G = (V, E)$, where $V$ is a set of vertices (or nodes) and $E$ is a set of edges (or links) that connect pairs of vertices. Each edge in $E$ is represented as a pair $(\nu, \mu)$ where $\nu$ and $\mu$ are vertices in $V$. Each node $\nu$ may have associated features, typically represented by a scalar or a vector $x_\nu$. Similarly, the edge $(\nu, \mu)$ connecting nodes $\nu$ and $\mu$ may also have vectorial or scalar features $e_{\mu,\nu}$, which may indicate properties such as the strength of the connection. One way to represent a graph is the adjacency matrix, i.e., a square matrix $A$ where the element $A_{\nu,\mu}$ indicates the presence (and possibly the weight) of an edge between vertex $\nu$ and vertex $\mu$. Mathematically,

$$A_{\nu,\mu} = \begin{cases} e_{\nu\mu} & \text{if } \nu \text{ and } \mu \text{ are linked} \\ 0 & \text{otherwise} \end{cases}$$

where $e_{\nu\mu} = 1$ if the graph is unweighted.

### 2.2 Graph neural networks

GNNs are a new type of neural network designed to process graph-based information directly, overcoming the limitations of traditional machine learning techniques. For example, using a multilayer perceptron (MLP) to analyze the adjacency matrix of a graph is inefficient because the performance of the MLP strongly depends on the arbitrary order of the nodes used in the adjacency matrix. In other words, this model is not invariant to permutations (Hamilton, 2020). Furthermore, even small changes, such as adding a node, require significant changes to the neural network. GNNs overcome these limitations by capturing the connectivity of the graph without being tied to a particular set of nodes by mapping the $G$ graph to a $\mathbb{R}^d$ vector space in which similar nodes are embedded close together. The final embedding results from a message passing, where each node in the graph communicates with its neighbors by exchanging information along the edges that connect them. Specifically, each node aggregates the information it receives from its first neighbors and updates its state based on these inputs. Messages are passed and aggregated in multiple iterations (multiple layers), so that the final message of each node contains information from nodes several steps away on the

graph. This allows GNN to collect and aggregate information from increasingly distant parts of the graph, thus capturing complex relationships within the graph structure. Mathematically, the entire message-passing process can be described as follows.

$$\begin{cases} h_\nu^{(0)} = x_\nu \\ h_\nu^{(l+1)} = \sigma\bigg(W_l \cdot AGG_l\Big(h_u^{(l)}, \ \forall \ u \in \mathcal{N}_\nu\Big) + B_l \cdot h_\nu^{(l)}\bigg) \\ z_\nu = h_\nu^{(L)} \end{cases}$$

$\forall l \in \{1, .., L-1\}$, where $x_\nu$ is the initial feature vector of $\nu$, $h_\nu^{(l)}$ is the $l-$embedding of $\nu$ ($l-$layer operation result on $\nu$) and $\sigma(\cdot)$ is a non linearity function. $W_l$ and $B_l$ are the $l-$elements of weight matrices. They do not depend on the specific node, so these values are shared between the nodes of a specific layer. $AGG_l(\cdot)$ is the aggregation operation. It is node order-invariant, such as the mean function, the maximum function, etc... $z_\nu$ is the final embedding of the node.

## 2.3 Proximal policy optimization

Proximal policy optimization is a popular deep reinforcement learning algorithm (Schulman et al., 2017) designed to achieve a favorable trade-off between stable learning and efficient data utilization. It belongs to the family of policy gradient techniques, which directly adjust a parameterized policy by estimating the gradient of the expected cumulative reward and updating the parameters via stochastic gradient ascent.

PPO is inspired by previous methods such as Trust Region Policy Optimization (TRPO) (Schulman et al., 2015a), which imposed explicit constraints to limit excessive policy changes during training. Although TRPO promotes stable learning through a strict trust region constraint, it is computationally demanding due to the need to solve a constrained optimization problem. To overcome this drawback, PPO replaces the hard constraint with a surrogate loss that implicitly limits policy updates, avoiding the use of second-order optimization methods. Specifically, this surrogate objective employs a clipping strategy that limits the deviation between the new and old policies by constraining the change in their probability ratio. Formally, the objective is defined as

$$\mathcal{L}^{actor} = \mathbb{E}\bigg[\min\Big(r_t(\theta)\,\hat{A}_t, \ clip\big(r_t(\theta), \, 1 - \epsilon, \, 1 + \epsilon\big)\hat{A}_t\Big)\bigg]$$

where $r_t(\theta) = \frac{\pi_\theta(a_t|s_t)}{\pi_{\theta_{old}}(a_t|s_t)}$, and $\hat{A}_t$ is an estimator of the advantage function at timestep t, which tells the agent how good an action at time $t$ is relative to its usual behavior.

In other words, when a policy update induces a large deviation from the previous policy, its effect is attenuated. This mechanism promotes stable training dynamics while still allowing for consistent performance gains.

The canonical PPO framework adopts an actor–critic architecture composed of two distinct neural networks. The actor, parameterized by $\theta$, represents the policy $\pi_\theta(\mathcal{A} \mid \mathcal{S})$, defining a probability distribution over actions conditioned on the current state. The critic, with parameters $\phi$, approximates the state-value function $V_\phi(S)$, which serves to estimate the expected cumulative reward from a given state under the current policy. In this work, we build on the implementations described in Huang et al. (2022), Engstrom et al. (2020), and Raffin et al. (2021), which incorporate several enhancements to the original PPO formulation. Among these extensions, the introduction of generalized advantage estimation (GAE) (Schulman et al., 2015b) is particularly notable. GAE achieves a favorable bias–variance trade-off by computing advantages as a weighted combination of temporally shifted temporal-difference residuals. As a result, it provides a more reliable and smoother approximation of the advantage function, contributing to improved stability during training. Below, a simplified version of the PPO pseudo-code is presented, highlighting the main components and operational steps of the algorithm.

1: Initialize actor network $\pi_{\theta_0}$
2: Initialize critic network $V_{\phi_0}$
3: **for** *iteration* $= 1$ **to** $N$ **do**
4:     Generate $\{s_0, a_0, r_0, \ldots, s_T, a_T, r_T\}$ following $\pi_{\theta_0}$
5:     **for** $t = 1$ **to** $T - 1$ **do**
6:         Compute $G_t = \sum_{k=t+1}^{T} \gamma^{k-t-1} r_k$
7:     **end for**
8:     Compute advantage estimation using GAE
9:     **for** *epoch* $= 1$ **to** $M$ **do**
10:         Compute $\mathcal{L}^{\text{tot}} = a\,\mathcal{L}^{actor} + b\,\mathcal{L}^{critic} - c\,\mathcal{L}^{entropy}$
11:         Update NNs using ADAM optimizer with $\nabla \mathcal{L}^{\text{tot}}$
12:     **end for**
13: **end for**

where $\mathcal{L}^{tot}$ denotes the overall objective function, consisting of a weighted combination of several components, including actor loss $\mathcal{L}^{actor}$, critic loss $\mathcal{L}^{critic}$, and the term of entropy regularization $\mathcal{L}^{entropy}$. The coefficients assigned to each component act as global hyper-parameters of the algorithm and can be adjusted according to the requirements of the specific task. In particular, the entropy term controls the degree of exploration. Higher entropy weights favor a broader exploration of the action space, while lower values favor faster convergence at the expense of potentially reduced robustness. For further details, see Huang et al. (2022).

## 3 Modeling of the problem and of the agent

This section is devoted to describing the formalization of the problem and the design of the GNN.

### 3.1 The delivery route problem as a Markov decision process

The problem we address in this work is a variant of the TSP (Goyal, 2010), one of the most notorious examples of NP-hard constrained combinatorial problems. The main objective of TSP is to pass through each available city (represented as a node) of a fully connected and weighted graph just once to minimize the total path. The TSP arises in many different contexts. Typical applications are vehicle routing (Braekers et al., 2016), clustering (Selim & Alsultan, 1991), and job scheduling (Błażewicz et al., 1996). The formulation as a TSP is essentially the simplest way to solve these problems. Most applications originate from real-world problems and thus seem to be of particular interest (Lenstra & Kan, 1975; Hahsler & Hornik, 2007).

In the routing problem we are interested in, the main differences from the TSP are that it is not necessary to pass through each node only once and that the graph is not necessarily fully connected. In fact, in real-world applications, the network map representing the city streets that the delivery driver must travel may consist of dead-end streets or geographic points that may not be directly accessible from one another except by revisiting other nodes.

We consider an environment obtained by randomly placing $N$ delivery destinations uniformly in a square plane of side length $L$. The resulting set of way-points is then connected by randomly assigning a fixed number of edges such that the corresponding graph results connected [1]. This construction yields a sparse, weighted graph that mimics realistic road networks. Formally, the environment is modeled as a weighted graph $G = (V, E, D)$, where $V = \{1, \ldots, N\}$ denotes the set of nodes, $E \subseteq V \times V$ is the set of admissible edges, and $D : E \to \mathbb{R}^+$ assigns a traversal cost to each edge. For each node $\nu \in V$, a characteristic vector $\vec{x}_\nu$ is stored, defined as

$$\vec{x}_\nu = \big(x_\nu, y_\nu, v_\nu, p_\nu, d_\nu\big),$$

where $(x_\nu, y_\nu) \in (0, L)^2$ are the spatial coordinates of the node, $v_\nu \in \{\pm 1\}$ encodes whether the node has been visited or not, $p_\nu \in \{0, 1\}$ indicates the presence of the traveler at the node, and $d_\nu$ is the degree

---

[1] A graph is connected if a vertex can be joined from any other vertex by following one or more edges.

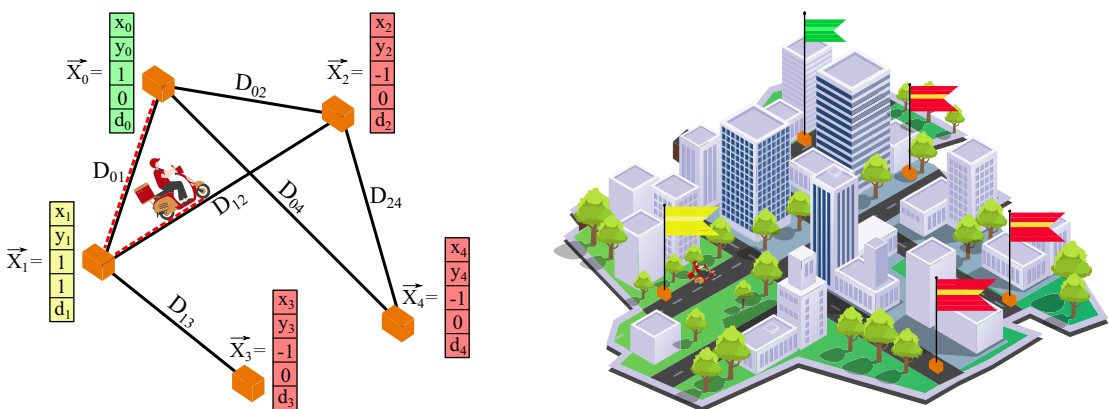

Figure 1: In this case, the nodes of the graph are the delivery targets, and all the connections between them represent the possible ways to reach each goal. The overall graph with edge and node information is an example of an observation of the environment. In each node $\nu$ a vector information $\vec{x}_\nu$ is stored, and it consists of the two coordinates of the position on the plane $(x_\nu, y_\nu)$, if the node has been visited or not $\{\pm 1\}$, the presence of the traveler $\{0, 1\}$ and the degree of the node $d_\nu$. The distance $D(\nu, \mu)$ between the destinations $\nu$ and $\mu$ is contained into the edge $(\nu, \mu)$ if the connection between $\nu$ and $\mu$ exists. Source: adapted from 3D City Map Vectors by Vecteezy.

of the node. If an edge $(\nu, \mu) \in E$ exists, it stores the Euclidean distance $D_{\nu\mu}$ between the corresponding way-points, as can be seen in Figure 1.

At the beginning of each GRL episode, both the graph topology and the delivery destinations coordinates are resampled, resulting in a dynamic environment in which the agent must generalize across different instances of the routing problem. To model the sequential decision-making process, we formulate the delivery routing task as a discrete-time Markov Decision Process (MDP) with a finite horizon. An MDP is characterized by the tuple

$$\mathcal{M} = (\mathcal{S}, \mathcal{A}, R, P),$$

where $\mathcal{S}$ denotes the set of possible states, $\mathcal{A}$ the set of admissible actions, $R$ the reward function, and $P$ the state transition kernel. The state of the environment is defined as the entire graph. Specifically, the state space $\mathcal{S}$ is the set of graphs that share the same topology as $G$, but differ in their nodes and edges features:

$$\mathcal{S} = \left\{ \mathcal{G} = (V, E, \mathbf{X}, \mathbf{D}) \mid \mathbf{X} = \{\vec{x}_\nu\}_{\nu \in V}, \mathbf{D} = \{d_{\nu\mu}\}_{(\nu,\mu) \in E} \right\} = \left\{ \mathcal{G} = (V, E, \mathbf{X}) \mid \mathbf{X} = \{\vec{x}_\nu\}_{\nu \in V} \right\}$$

The Markov property holds, as the complete configuration of the environment, including the position of the driver and the visited nodes, is fully encoded in the node feature matrix $\mathbf{X}$. In fact, the matrix $\mathbf{D}$ of the weights associated with the edges is fully determined by the coordinates of the positions of the nodes encoded in $\mathbf{X}$.

The action space is dynamic and depends on the current position of the traveler. In a given state $\mathcal{G}_t$, the set of admissible actions is

$$\mathcal{A}(\mathcal{G}_t) = \{\nu' \in V \mid (\nu_t, \nu') \in E\},$$

where $\nu_t$ denotes the node currently occupied by the traveler. Invalid actions are removed through an action-masking mechanism (Huang & Ontañón, 2020), which prevents the agent from selecting transitions that violate the graph topology. The transition dynamics is deterministic. The selection of the action $a_t = \nu'$ results in an updated graph state $\mathcal{G}_{t+1}$, obtained by moving the traveler to node $\nu'$ and therefore updating the parts of the feature matrix $\mathbf{X}$ corresponding to $v_\nu$, $p_\nu$, $v_{\nu'}$ $p_{\nu'}$.

$$P(\mathcal{G}_{t+1} \mid \mathcal{G}_t, a_t) = 1, \quad \mathbf{X}_t = \begin{pmatrix} \cdot & \cdot\cdot & \cdots & \cdot\cdot & \cdot \\ x_\nu & y_\nu & 1 & 1 & d_\nu \\ x_{\nu'} & y_{\nu'} & -1 & 0 & d_{\nu'} \\ \cdot & \cdot\cdot & \cdots & \cdot\cdot & \cdot \end{pmatrix} \rightarrow \mathbf{X}_{t+1} = \begin{pmatrix} \cdot & \cdot\cdot & \cdots & \cdot\cdot & \cdot \\ x_\nu & y_\nu & 1 & 0 & d_\nu \\ x_{\nu'} & y_{\nu'} & 1 & 1 & d_{\nu'} \\ \cdot & \cdot\cdot & \cdots & \cdot\cdot & \cdot \end{pmatrix}$$

As can be seen in the transition $\mathbf{X}_t \to \mathbf{X}_{t+1}$ and in Figure 1, the characteristic corresponding to the presence of the driver in the city $p_\nu$ is updated from 1 to 0 because the driver has left node $\nu$. At the same time, $v_\nu$, which indicates whether node $\nu$ has been visited or not, does not change its value because the presence of the traveler is sufficient for the node to be marked as already visited. The node $\nu'$, on the other hand, after the transition from $t$ to $t+1$, changes its tuple $p_{\nu'}$, $v_{\nu'}$ from $(-1, 0)$ to $(1, 1)$ because the node is visited and the driver is placed on the node $\nu'$ at time $t+1$.

Unlike the classical TSP, no constraint is imposed on visiting each node only once, allowing the agent to traverse the same way-point multiple times if necessary. An episode ends when all delivery destinations have been visited or when a maximum number of steps $N_{\max} = \frac{1}{2}N(N-1)$ is reached, corresponding to the number of edges in a fully connected graph. As the cumulative path distance increases with the length of the episode, effective learning is expected to gradually reduce the number of steps required to complete the task.

The reward function is designed to be scale invariant with respect to the size of the environment. Since the node coordinates $(x_\nu, y_\nu)$ are uniformly sampled as $x_\nu \sim \mathcal{U}(0, L)$, $y_\nu \sim \mathcal{U}(0, L)$, the maximum possible distance in the plane is $\sqrt{2}L$, which is used to normalize the edge costs. The immediate reward associated with moving from node $\nu$ to node $\mu$ is defined as

$$r(\nu \to \mu) = -\frac{D_{\nu\mu}}{\sqrt{2}L},$$

so that $r(\nu \to \mu) \in (-1, 0)$. With a discount factor $\gamma = 1$, the return over an episode of $N_{\text{steps}}$ is

$$R = \sum_{l=1}^{N_{\text{steps}}} r_l = -\frac{1}{\sqrt{2}L} \sum_{(\nu,\mu)} D_{\nu\mu}, \qquad -N_{\text{steps}} < R < 0.$$

To make the return independent of the number of nodes, we introduce a normalization factor $f(N)$ and define

$$R = -\frac{1}{f(N)\sqrt{2}L} \sum_{(\nu,\mu)} D_{\nu\mu}, \qquad -1 < R < 0.$$

By imposing the condition $L \cdot f(N) = K^{-1}$, where $K > 0$ is a tunable constant, the reward function can be rewritten as

$$r(\mathcal{G}_t \mid a_t = \nu \to \mu) = -\frac{K}{\sqrt{2}} D_{\nu\mu}.$$

This choice ensures that the return function remains normalized regardless of the graph topology with a single hyperparameter choice. The reward is always negative, so the goal of the agent is to learn to complete the episode as quickly as possible and then gradually refine its strategy to decrease the length of the path followed. In such a scenario, the optimal policy $\pi^* : \mathcal{S} \to \mathcal{A}$ maximizes the expected cumulative reward:

$$\pi^* = \arg\max_\pi \ \mathbb{E}_\pi \left[ \sum_{t=0}^{N_{steps}} r(\mathcal{G}_t \mid a_t) \right],$$

subject to the connectivity constraints imposed by $E$.

## 3.2 Agent

As mentioned in the PPO section, the brain of the agent is based on two components, which are the critic network and the actor network. In this case, both are graph convolutional neural networks, and specifically, each convolutional layer is implemented as described in Morris et al. (2019). This choice is motivated by the fact that, in the aggregation function, the easiest way to account for edge information is to weight node features with it:

$$h_\nu^{(l+1)} = \tanh\left( W_l \cdot \sum_{u \in \mathcal{N}_\nu} e_{\nu,u} \cdot h_u^{(l)} + B_l \cdot h_\nu^{(l)} \right), \ l = 0, 1$$

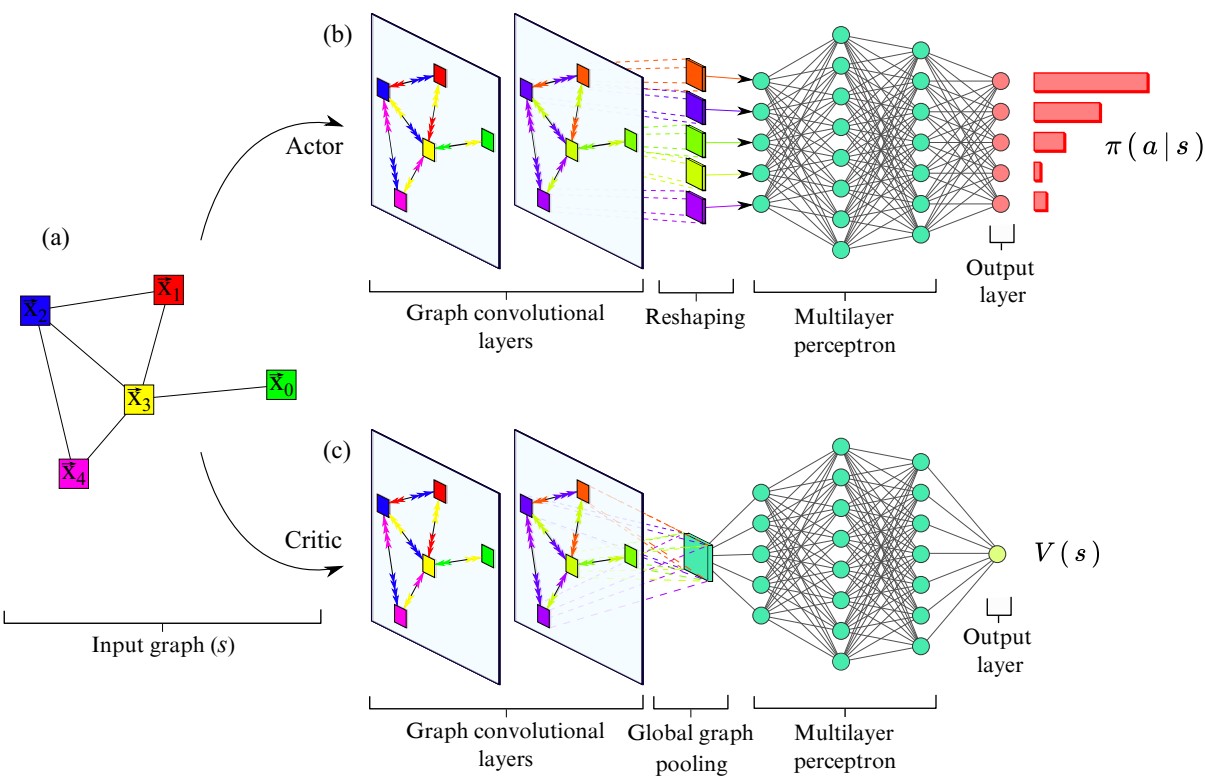

Figure 2: The diagram illustrates the structure of the implemented GNN model. The architecture is segmented into three parts, labeled (a), (b), and (c): (a) The input to the GNN is a graph $s$, where $\{\vec{x}_0, \vec{x}_1, \vec{x}_2, \vec{x}_3, \vec{x}_4\}$ represent features associated with different delivery targets, and edges denote the connecting roads. (b) The actor network represents the policy network in the PPO, which is responsible for determining the actions taken by the agent. It uses two convolutional graph layers to aggregate features of nodes based on their neighborhood structures. The output is reshaped to go through a Multilayer Perceptron (MLP) that produces a distribution of policies over possible actions $\pi(a|s)$. (c) The critic network estimates the value function, which evaluates the quality of the actions chosen by the actor. Like the actor network, the critic also uses graph convolutional layers to process the input graph, followed by a global graph pooling operation that aggregates node-level information into a single global graph representation. This global feature vector is then processed by an MLP, which outputs the value estimate $V(s)$, a scalar that represents the expected future rewards for the current state and action.

where $e_{\nu,u} \in \mathbb{R}$. The two GNNs take the graph state $\mathcal{G}_t$ as input, and after two convolutional layers, the two architectures differ. In the case of the critic network, a pooling of the global average is performed before an MLP, as shown in Figure 2. The output of this network is a scalar value consisting of the value function $V(s)$. The actor network, on the other hand, creates a reshaping of the input vector given by the graph convolution filter. Then it is fed into the final MLP, and the output of the overall architecture is a probabilistic distribution of all possible actions $\pi(\mathcal{A}|s)$.

## 4 Testing and results

In this Section, we show all the results of the testing part of the algorithm implemented for the delivery path problem. To do so, we specify that all simulations were performed on an AMD EPYC 7352 hardware supported by an NVIDIA RTX A6000 GPU. The entire software was written on Pytorch (Paszke et al., 2019), PyGeometric (Fey & Lenssen, 2019), and NetworkX (Hagberg et al., 2008). Our goal is to study the delivery path problem in instances that deviate significantly from fully connected graphs. Therefore,

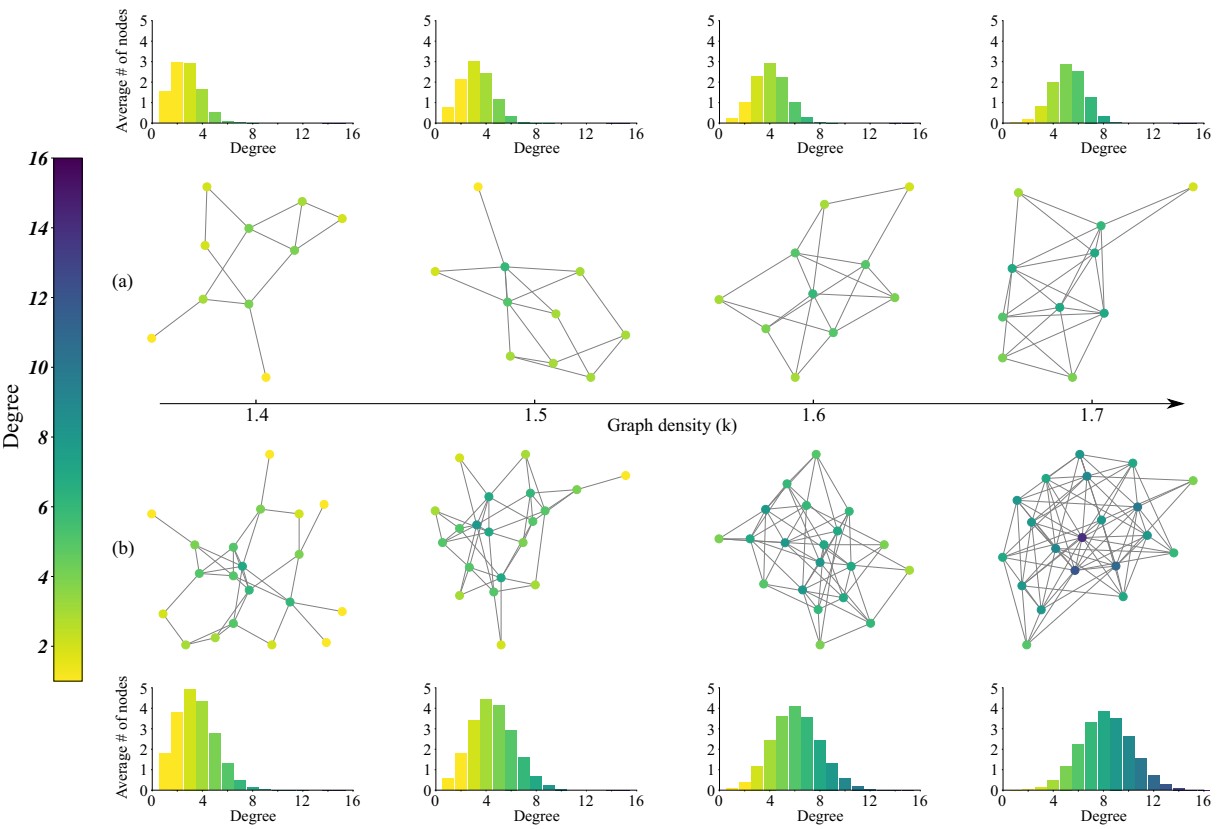

Figure 3: Histogram and associated example of a sampled graph for each exponent $k$ (ranging from 1.4 to 1.7) of the power law representing the density of the graph and for instances $N = 10, 20$. The $N = 10$ case (row a) is illustrated at the top, while the $N = 20$ instance (row b) is provided downward. Each histogram represents a degree distribution in different connectivity regimes, where the degree, i.e., the number of connections per node, is shown on the x-axis, while the average number of nodes with a given degree is on the y-axis. As $k$ decreases, for both $N$ cases, the degree distributions shift, capturing the transition from highly connected graphs - in which each node has at least two neighbors - to more sparse graphs, in which nodes with only one neighbor begin to emerge.

it is essential to identify a qualitative and reasonable sparsity regime that is meaningful and relevant to this analysis. There are several metrics to measure the sparsity. Examples are the Gini index and the edge density $\rho$ (Goswami et al., 2018). In this work, we refer to the latter. The edge density of a simple undirected graph $G = (V, E)$ is given by

$$\rho(N) = |E| / \binom{|V|}{2}$$

If $\rho(N) \to 0$ for the number of nodes $N \to \infty$ the graph is sparse defined. This implies that any power law $|E| \sim o(N^2)$ falls within this regime. We focus on four specific cases: $|E| = \lfloor \frac{1}{2} N^k \rfloor + 1$ where $k = 1.4, 1.5, 1.6, 1.7$. The motivation for focusing on these values becomes clear when observing the degree distributions shown in Figure 3. Starting from the case $k = 1.7$ and moving to $k = 1.4$, we notice that the probability of obtaining a graph where some nodes have only one connection steadily increases. This range is particularly insightful, as it allows us to study the transition from graphs where each node has at least two neighbors to scenarios where nodes with only one neighbor start to appear.

Based on this observation, the following statistical analysis investigates the learning performance of the algorithm under these varying connectivity conditions. The study includes 10 simulations for graphs with $N = 10$ and $N = 20$ nodes with fixed connectivity and an example with 30 nodes without statistical analysis.

Table 1: The main hyper-parameters used for the simulations are listed in the table. In particular, the learning rate, the number of hidden neurons per layer, the total number of time steps required for each simulation, and the level of entropy are given.

| Nodes | Hyper-parameters | k Values | | | |
|---|---|---|---|---|---|
| | | $k = 1.4$ | $k = 1.5$ | $k = 1.6$ | $k = 1.7$ |
| 10 | Learning Rate | $5e-05$ | $5e-05$ | $5e-05$ | $1e-04$ |
| | Hidden Neurons | 128 | 128 | 128 | 128 |
| | Total time-steps | $5e+05$ | $5e+05$ | $5e+05$ | $5e+05$ |
| | Entropy coefficient | 0.01 | 0.01 | 0.01 | 0.01 |
| 20 | Learning Rate | $1e-05$ | $1e-05$ | $5e-05$ | $5e-05$ |
| | Hidden Neurons | 256 | 256 | 128 | 128 |
| | Total time-steps | $1e+06$ | $1e+06$ | $1e+06$ | $1e+06$ |
| | Entropy coefficient | 0.1 | 0.1 | 0.01 | 0.05 |
| 30 | Learning Rate | $1e-05$ | $1e-05$ | $1e-05$ | $1e-05$ |
| | Hidden Neurons | 512 | 512 | 512 | 512 |
| | Total time-steps | $2e+06$ | $2e+06$ | $2e+06$ | $2e+06$ |
| | Entropy coefficient | 0.01 | 0.05 | 0.05 | 0.05 |

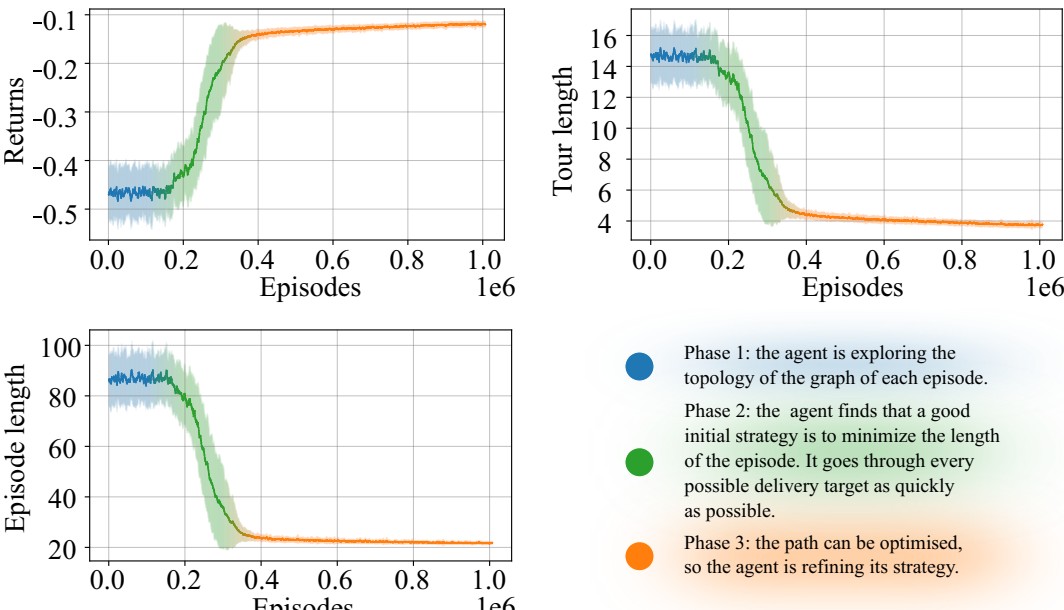

Figure 4: Average performance over 10 runs of the agent, highlighting three stages of learning. As episodes progress, the agent reduces episode length, increases cumulative rewards (returns), and optimizes tour length. The shaded areas represent the variability between executions, and the phases describe the progression of the agent from exploring the graph to minimizing the episode length and finally to refining the strategy to optimize the tour.

Each simulation collects a set of information about the learning trend, which are the cumulative reward (i.e. the return), the episode length (i.e. the number of steps required by the agent to achieve a completed episode), and the length of the tour, which is the cumulative sum of each distance covered by the driver. Each of these quantities in the simulations is represented as a function of the number of episodes, while the error bars represent the standard deviation from the mean, highlighting the variability of the results. A 10-point convolution was applied to smooth the curves to reduce noise and present a clearer trend.

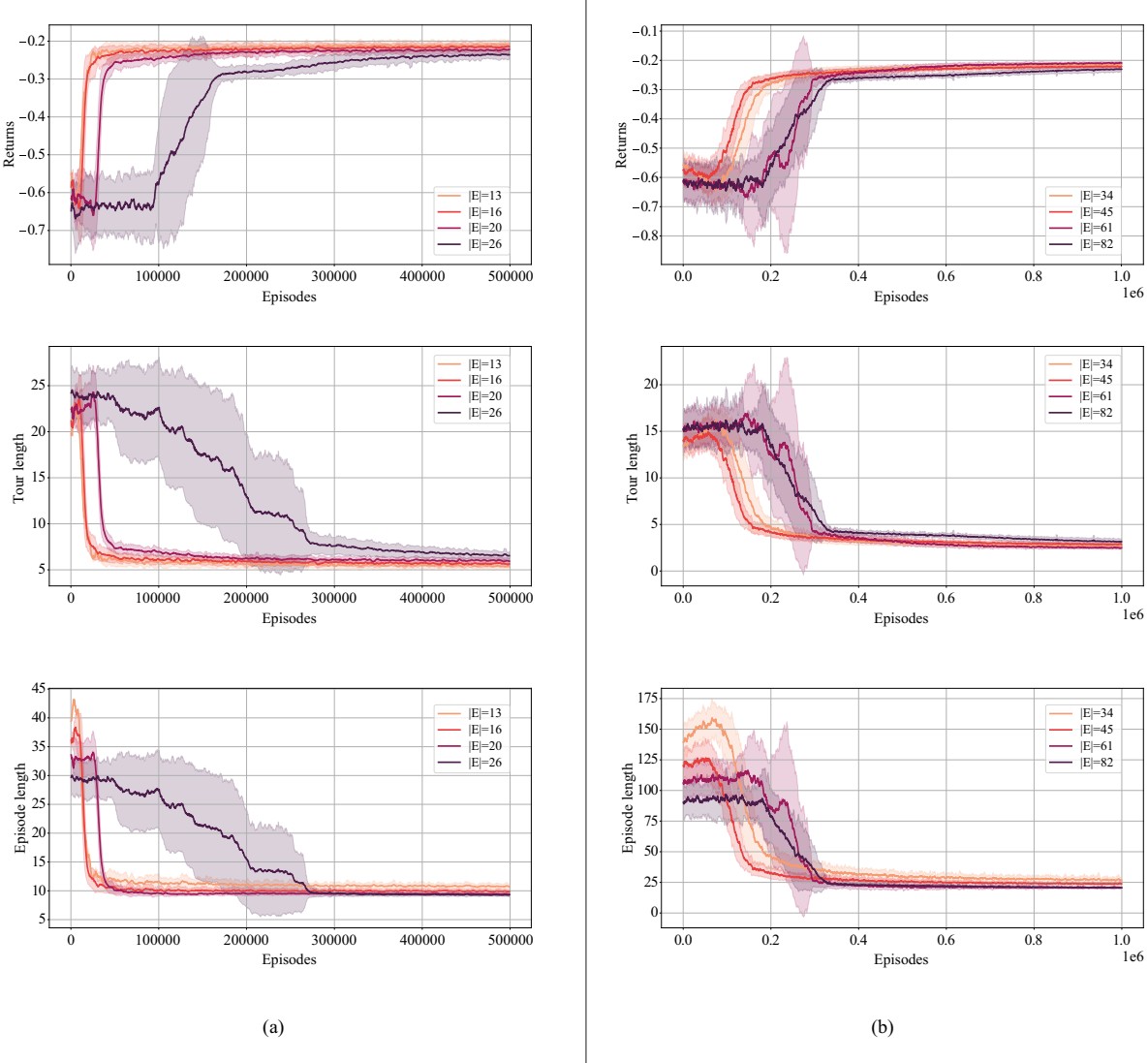

(a)                                        (b)

Figure 5: Here, we show simulation results for the delivery path problem addressed for 10 (column (a)) and 20 (column (b)) nodes. For both return functions, the tour length and episode length are given as the result of averaging over 10 executions. As illustrated in the figure, the algorithm is able to learn a good strategy to solve the problem. In fact, the architecture is able to refine its strategy to reduce the length of the tour. Obviously, there are differences between simulations for different values of $k$, on which the sparsity of the graph depends. Identifying a strategy that seeks a path in which each delivery target is reached the minimum number of times is easier for a sparser graph. For the denser case, however, this task is more difficult. Therefore, a more extensive initial exploration phase is required.

Figure 4 illustrates an example of the average behavior of 10 simulations, where the agent goes through three phases of learning and optimization. The x-axis represents the number of episodes, while the y-axes of the three graphs display different performance metrics. The top left picture shows the cumulative reward

trend, reflecting the increasing ability of the agent to achieve higher rewards as it refines its strategy over episodes. The lower left figure depicts how the length of the episode decreases over time, indicating the growing efficiency of the agent in completing episodes as learning progresses. The top left image reveals the initially high tour length being optimized across episodes, demonstrating the advancement of the agent in minimizing the path followed.

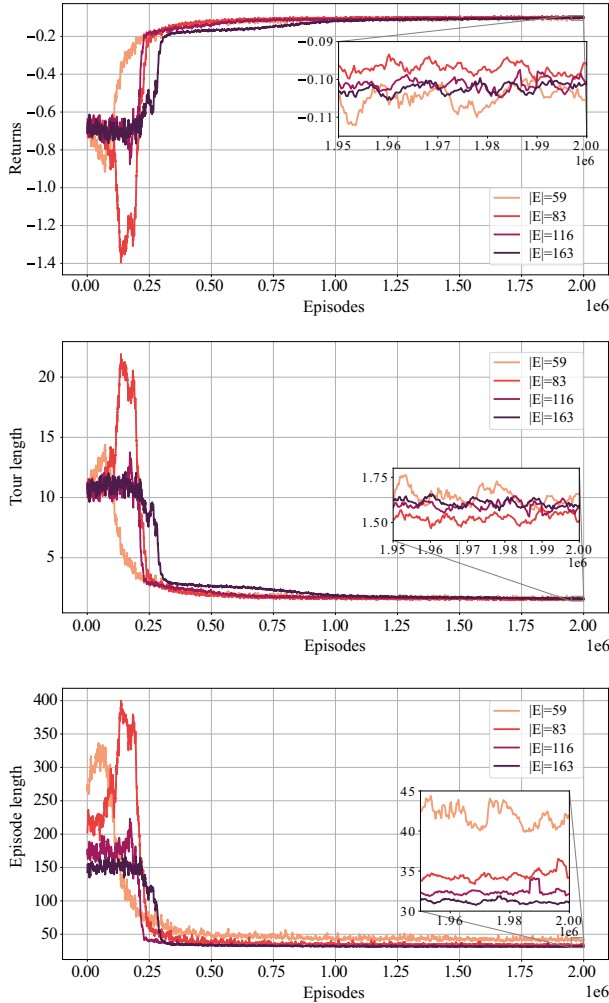

Figure 6: Simulation results for 30 nodes instance of the delivery path problem . The return function, path length and episode length are given. Although, due to the lack of statistical analysis, we cannot speak of definitive results, the algorithm seems to be able to learn a good strategy to solve the problem. As before, the architecture is able to refine its strategy further to reduce the tour length as a function of the $k$ sparsity level. Of course, we expect that in a sparser regime, the episode length is higher at the end of learning because the traveler encounters more delivery targets where there is only one link to reach them than in the denser case.

The shaded areas around each curve represent the standard deviation, capturing the variability among the 10 simulations. Three phases of the agent's behavior can be identified. During phase 1, the agent explores the topology of the graph. In phase 2, it adopts a strategy to minimize the duration of the episode by quickly visiting all delivery targets. Phase 3 represents the refinement of the strategy to optimize the path length. Following such a scheme, we compare all simulations between different sparsity regimes, as shown in Figure 5. In all conditions, the agent can find a good strategy to solve the problem. This is because, in both the 10 and 20 cases, the final episode length is observed to be very close to the minimum number of delivery targets

present on the graph. In addition, the length of the trip tends to decrease consistently until the end of the training. Note that there is no guaranty of the optimality of such solutions. However, the number of possible paths leading to the solution is higher for problem instances with higher graph density. Therefore, we can appreciate that, in such cases, the agent needs a longer exploration phase of the space of possible solutions in the first training phase. As soon as the agent learns that a first large gain in terms of the reward is given by following paths that pass through each city the least number of times, the trend shows a significant jump.

The third part of the training is clearly characterized by the refinishing, and consequently the minimization, of the path length. We chose to end the learning phase at $5 \cdot 10^5$ iterations for the case of 10 nodes and at $1 \cdot 10^6$ iterations for the case of 20 nodes. However, this decision could be modified to obtain a more fully trained GNN. In particular, for the cases with 20 nodes with $k = 1.7$, the upward trend observed in the final performance suggests that further training could produce further improvements. In the specific case of 20 nodes and 61 edges, instead, we notice a particularly high-performing behavior that can be seen as a special coincidence of having made a very good choice in the space of hyperparameter.

For this analysis, the list of hyper-parameters used is given in Table 1. Among them, the entropy coefficient is particularly relevant because it is related to the ability of the agent to explore the graph. By setting this coefficient too low, the traveler tends to self-locate in a portion of the graph without exploring it at all. This effect is particularly noticeable when going from 10 to 20 nodes with a low sparsity level, especially for $k = 1.4$ and $k = 1.5$.

In the end, in Figure 6 we show a single instance execution with 30 nodes, where it is evident that the agent, as before, is able to recognize a good strategy to solve the problem for all $k$-sparsity regimes. To do this, the neural network was modified with a larger number of neurons per layer, up to 512. It is natural to believe that as the complexity of the problem increases, the GNN must also increase its complexity to solve it. The duration of training must also increase to obtain sufficiently good solutions. In principle, therefore, we can assume that far more complex instances of this problem can be solved by making use of more computational resources and attempting to carefully optimize both the neural network parameters and the algorithm hyper-parameters.

## 5   Discussion and conclusions

This study explores how varying the sparsity of a graph representing the state of the environment affects the combination of PPO and GNNs techniques. As testbed, we focus on the problem of delivering goods on multiple locations naturally defined on sparse graphs. Our experiments with the GRL framework show that GRL-based heuristics are effective and robust in addressing this problem across a transition of four different levels of graph sparsity, each characterized by the exponent of a power-law distribution. These scenarios range from graphs with numerous low-degree nodes, which impose inefficient visiting paths, to highly connected topologies that require extensive exploration due to the multiple alternatives available. To allow meaningful comparisons across different graph topologies, we introduce a method to normalize the return function by involving the length of the GRL episode and the number of graph nodes. Such normalization allows us to adjust just one parameter in order to make the cumulative reward scalable with respect to the scale and topology of the graph. We perform a statistical analysis of performance for graphs with up to 20 nodes. In addiction, we illustrate the results using an example from a single run on a 30-node graph. The agent shows good training performances, demonstrating the ability to identify efficient solutions in terms of path length, the average duration of a GRL episode, and the return achieved. Our results validate the approach and highlight its potential for studying graph NP-hard problems in GRL for different connectivity levels. The study also acknowledges the limitations of the proposed method, in particular its inability to guaranty the absolute optimal solution, despite consistently producing high-quality results. Future research could focus on training the GNN in specific cases of the graph-state with the aim of solving problems with a larger number of nodes but with the same connectivity patterns as the training graphs. Overall, this research provides an analysis that may be useful for further addressing other graph-based problems in complex, real-world settings.

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
