# OpenReview forum: "Graph reinforcement learning resistant to sparsity scaling"
_TMLR — Rejected by TMLR_

### Review · Reviewer_R6UR · 2026-02-15

**Summary Of Contributions:**

The authors investigate applying RL to a variation of the Traveling Salesman Problem (TSP).
Their variation of TSP operates on non-fully connected graphs and allows re-visiting nodes. They explore different network sizes and edge densities and show that the agent is successfully able to learn a good policy in all cases.

As policy and VF, they use GNNs. In the case of the policy, the output of the GNN is flattened and fed through an MLP to predict the action probabilities.

**Audience:**

No

**Audience Explanation:**

While the topic itself might receive some interest, there are several issues with the paper that limit its utility for the reader.

# Policy choice.

While the paper positions itself as using a Graph Neural Network as policy and value function, *it flattens the outputs of the GNN to predict action probabilities using a MLP*. This architecture choice is unexplained and, in my opinion, negates the majority of benefits of the GNN, primarily that it removes any invariance from the architecture (for example, the same graph, but rotated, would result in different answers in the current architecture). It is unclear to me why the policy was not chosen to a GNN "end to end".

# No evaluation of quality of results

While the policy is shown to learn some decent policy, it is not compared to what the "optimal" solution would have been on a given Topology, so it is hard to judge how well the approach really works.

# No generalization experiments

A great feature of GNNs is that they can generalize to a different number of node. Hence, it would have been interesting to see how well policies can generalize across graph sizes and densities. (However, given the MLP in the policy, this is currently not possible).
At the moment, each policy is trained only for a given graph size and density, which I believe is unrealistic to be useful in practise. I would expect that for a given real world application, both of these variables are unknown (and changing), so a policy would need to generalize.

# The choice of reward is odd

The paper starts by motivating a reward/cost that is scaled by the size of the plan on which the graph is located. However, it then introduces a normalization function that then cancels this scaling, effectively reducing the reward to a simple edge distance times a manual hyperparameter. This doesn't seem like an optimal choice for a reward. Furthermore, why was the length scaling introduced  if it's removed again in the next step?

**Claims And Evidence:**

Yes

**Claims Explanation:**

They core of the paper is successful training runs on variations of graph size $N\in\{10, 20, 30\}$  and network densities.
The setup, approach, experiments and results are very clearly explained.

**Requested Changes:**

* Either justify the MLP in the actor architecture or run a pure GNN policy
* Compare results to the optimal path on each topology - how good are the policies really?
* Run generalization experiments across different graph sizes and densities
* Better justify your reward choice and address my comment on introduced and then removing the scaling by $L$.

---

### Review · Reviewer_326r · 2026-02-21

**Summary Of Contributions:**

The paper applies PPO with graph convolutional networks to a delivery routing problem on sparse random graphs. Sparsity is controlled via a power-law edge count $|E| = \lfloor \frac{1}{2}N^k \rfloor + 1$ for $k \in \{1.4, 1.5, 1.6, 1.7\}$, and experiments are conducted on graphs with 10, 20, and (a single instance of) 30 nodes. A return normalization is proposed for cross-topology comparison, and the main claim is that learning remains stable across these sparsity regimes.

The motivation -- studying routing on sparse rather than fully connected graphs --  is reasonable and practically relevant. The three-phase learning characterization (exploration -> episode minimization -> path refinement) is intuitive and clearly presented.

However, there are fundamental issues with the evaluation. There are no baselines whatsoever: no heuristics, no exact solver, no competing neural methods. At $N=20$, exact solvers would find optimal solutions in milliseconds. Hyperparameters are tuned per $(N,k)$ configuration, undermining the robustness claim. The problem itself reduces straightforwardly to an asymmetric TSP via shortest-path distances on the graph, which is not discussed. Overall, there is no clear insight or lesson learned that I believe would be of value to the community.

**Audience:**

No

**Audience Explanation:**

How graph structure and sparsity affect reinforcement learning for combinatorial optimization is a relevant topic. However, this particular paper does not produce findings that advance the community's understanding in a meaningful way.

The practical takeaway is that PPO+GCNN can learn on small, sparse graphs given sufficient per-instance hyperparameter tuning. This is not surprising and does not inform the design of methods for realistic problem sizes. Without baselines, the results cannot be contextualized against known alternatives. Without scale, they do not generalize. The paper does not identify any structural insight about how sparsity affects learning dynamics that a reader could transfer to other problems or architectures. There is also no discussion of failure modes, scaling limitations, or what makes certain sparsity regimes harder than others beyond surface-level observations.

In short, the paper does not offer a lesson that the community can build on.

**Claims And Evidence:**

No

**Claims Explanation:**

The central claim, i.e. that the method is "resistant to sparsity scaling", is not convincingly supported, for several reasons.

- **No baselines.** This is the most critical issue. The paper compares the agent only against itself across sparsity levels. At $N=20$, an exact solver finds the optimum in milliseconds, and even a simple nearest-neighbor heuristic would provide a useful reference point. Without any comparison, we cannot tell whether the learned policies are good, mediocre, or poor in absolute terms. The authors acknowledge there is "no guarantee of optimality" but make no effort to measure the gap.

- **Trivial problem scale.** The largest instances with statistical analysis have 20 nodes; the 30-node case is a single run. This is far too small to draw conclusions about scaling behavior. The community has been working at scales of hundreds to thousands of nodes for years. The title promises resistance to "sparsity scaling," but no actual scaling study is presented.

- **Per-instance hyperparameter tuning.** Table 1 shows that learning rate, hidden layer size, and entropy coefficient all vary across $(N,k)$ configurations. The authors themselves attribute one particularly strong result ($N=20$, $|E|=61$) to a lucky choice of hyperparameters. If different sparsity regimes require different hyperparameters, the method is not robust to sparsity, and it can be made to work with careful tuning, which is a much weaker statement.

- **Overstated contributions.** The environment is described as "dynamic," but within each episode, it is entirely static and deterministic. Resampling graph instances across episodes is a standard training procedure, not a dynamic environment in the RL sense. A part of the return normalization is presented as a contribution, but it amounts to a simple rescaling that does not go beyond what Kool et al. (2018) already employ: why not simply scale instances between $[0, 1]$ instead of having $[0, L]$ with variable $L$ ? The paper also states that "GNNs are a new type of neural network," which is inaccurate: GNNs have been studied for around a decade.

- **Missing problem reduction.** In my understanding, this delivery problem on a sparse graph can be reduced to an asymmetric TSP where inter-city distances are shortest-path distances on the graph. This is a classical and well-understood reduction that is never mentioned. It raises the question of what the GRL approach offers beyond solving an ATSP with standard tools, including, say, MatNet [1].

- **Redundant evaluation metrics.** Tour length, episode length, and returns are highly correlated quantities. Reporting all three across every configuration inflates the apparent thoroughness of the evaluation without adding genuine insight. Returns alone (or returns plus one secondary metric) would suffice.


---

[1] Kwon, Yeong-Dae, et al. "Matrix encoding networks for neural combinatorial optimization." Advances in Neural Information Processing Systems 34 (2021): 5138-5149.

**Requested Changes:**

**Critical (required for acceptance):**

1. **Add baselines.** Compare against (a) an exact solver (this is trivial at $N=20$), (b) classical heuristics (nearest neighbor, 2-opt adapted to sparse graphs), and (c) at least one neural baseline such as the attention model of Kool et al. (2018). Report optimality gaps for all methods.

2. **Scale up the experiments.** Extend to at least $N=50$–$100$ with proper statistical analysis (multiple runs). The current scale is insufficient to support any claim about scaling behavior.

3. **Fix hyperparameters across sparsity levels.** To genuinely test robustness, use a single set of hyperparameters across all $k$ values for a given $N$. The current per-configuration tuning invalidates the robustness claim.

4. **Evaluate on held-out test instances.** All current results are training curves. Report test-time performance on unseen graph instances to demonstrate generalization rather than just training convergence.

5. **Discuss the ATSP reduction.** Acknowledge that the problem reduces to an asymmetric TSP via shortest-path precomputation, and articulate clearly what the GRL approach offers beyond this well-known formulation. If my understanding of the problem is incorrect, I invite the authors to clearly explain why it cannot be reduced to a known one with publicly available baselines.

6. **Correct the "dynamic environment" framing.** The environment is static within each episode. Resampling across episodes is a standard instance generation, not a dynamic environment. This should be stated accurately.

**Recommended (would strengthen the paper):**

7. **Clarify evaluation metrics.** Returns alone capture the essential learning signal; tour length and episode length are derivable and redundant. Some metric, such as the optimality gap, would be much more informative.

8. **Improve Figure 3.** Use examples drawn from the same instance for each $k$ to enable direct visual comparison, and increase the size of the histogram insets.

9. **Report computational costs.** Wall-clock training times would help readers assess practical feasibility (which I believe is not a major issue given the small scales) and compare against classical solvers.

10. **Articulate a concrete takeaway.** What should a reader learn from this paper about the interaction between sparsity and GRL that they did not know before? Currently, this is missing.

---

### Review · Reviewer_WLdQ · 2026-02-23

**Summary Of Contributions:**

This paper considers a variant of the TSP in which nodes can be visited more than once and nodes are positioned in physical space on a graph that is not necessarily connected. The problem is approached with graph reinforcement learning through actor and critic networks parameterised by graph convolutional layers that are trained with PPO. The authors propose the use of reward function that normalises for the number of edges, which they suggest alleviates issues with poor performance of existing methods on graphs of varying density. Experiments on synthetic graphs of up to 30 nodes are presented.

**Audience:**

No

**Audience Explanation:**

I consider that the work has significant shortcomings in soundness, originality, clarity, and potential impact.

**Claims And Evidence:**

No

**Claims Explanation:**

I believe that the claims made in the paper are not appropriately supported by evidence, as I detail in comments C1-C5 below.

**Requested Changes:**

I hope that the questions and suggestions below will help the authors refine and improve their manuscript; however, as it currently stands, it is well below the standards of TMLR as a venue, and a single revision cycle is insufficient to address the significant issues with this manuscript.

C1. Missing justification for use of Graph RL: why is this approach necesary? It appears to me that the problem can be formulated as a (mixed) integer linear programming problem and solved exactly on instances significantly larger than 30 nodes and in a much shorter time. Therefore, I do not see any arguments for the use of this type of method. One could be that the constraints, rewards, etc. of the problem make such a framework inapplicable (not the case here). Another could be that the method scales to larger graphs while retaining good performance (also not the case here). See for example https://openreview.net/forum?id=HduK51xNtS, a Graph RL survey published in TMLR that should probably also be cited.

C2. The authors imply this problem is NP-hard but do not provide a proof or cite previous work that show this.

C3. The assertion that "such topology reflects the real world delivery networks" is clearly unfounded and needs scientific evidence.

C4. The authors also seem to imply that GRL methods struggle with this problem. However, the approach they propose is effectively standard with the exception of the normalised reward function. This is also a standard practice and amounts to changing the problem formulation. Evidence should be provided that, with this same problem definition, existing approaches struggle.

C5. The authors use separate hyperparameters and separately trained models for each level of graph density (Table 1). If my understanding is correct, this means that the model is not exposed to graphs of different densities during the training process, which makes the reward normalisation scheme entirely redundant.

C6. The state representation is lossy: states in which nodes have been visited more than once are collapsed to the same representation despite the fact that visiting the same nodes >1 one time incur additional costs. Why not simply use the node visit count instead of a boolean flag?

C7. The architecture is not applicable to graph instances of different sizes, which is a key desirable factor for graph reinforcement learning. Training the policy on small graphs and applying it on large graphs can give an advantage in terms of scalability to large instances compared to traditional methods. I would suggest exploring this aspect -- see the architectures in the survey I referenced above, for example.

C8. There are no baselines of any kind considered in the evaluation.

C9. Some suggestions on writing:
- Second paragraph of the introduction is much too long and should be broken up.
- 2.1: Are edge weights vectors or scalars? The adjacency matrix implies they can only be scalar in your formulation; be precise.
- You should also define the critic and entropy losses since the actor one is defined.
- Notation: $G$ denotes both return and graph, reconcile them.
- $\pi(\mathcal{A}|\mathcal{S})$ should be $\pi(a|s)$
- TSP is referred to as "notorious"; avoid phrases such as "brain of the agent"

---

### Decision · Action_Editor_nH9e · 2026-03-20

**Recommendation:** Reject

**Audience:**

No

**Audience Explanation:**

The experimental study is not convincing.

**Claims And Evidence:**

No

**Claims Explanation:**

All reviewers raised several issues with the paper, including the empirical study. The author did not provide a rebuttal.